# Interleukin-8 in Colorectal Cancer: A Systematic Review and Meta-Analysis of Its Potential Role as a Prognostic Biomarker

**DOI:** 10.3390/biomedicines10102631

**Published:** 2022-10-19

**Authors:** Chiara Bazzichetto, Michele Milella, Ilaria Zampiva, Francesca Simionato, Carla Azzurra Amoreo, Simonetta Buglioni, Chiara Pacelli, Loredana Le Pera, Teresa Colombo, Emilio Bria, Massimo Zeuli, Donatella Del Bufalo, Isabella Sperduti, Fabiana Conciatori

**Affiliations:** 1Preclinical Models and New Therapeutic Agents Unit, IRCCS Regina Elena National Cancer Institute, 00144 Rome, Italy; 2Medical Oncology 1, IRCCS Regina Elena National Cancer Institute, 00144 Rome, Italy; 3Section of Oncology, Department of Medicine, University of Verona-School of Medicine and Verona University Hospital Trust, 37134 Verona, Italy; 4Department of Oncology, San Bortolo General Hospital, 36100 Vicenza, Italy; 5Pathology Unit, IRCCS Regina Elena National Cancer Institute, 00144 Rome, Italy; 6Department of Biochemical Sciences “A. Rossi Fanelli”, Sapienza University of Rome, 00185 Rome, Italy; 7Servizio Grandi Strumentazioni e Core Facilities, Istituto Superiore di Sanità (ISS), 00161 Rome, Italy; 8Institute of Molecular Biology and Pathology-National Research Council (IBPM-CNR), 00185 Rome, Italy; 9Medical Oncology, Fondazione Policlinico Universitario Agostino Gemelli IRCCS, Università Cattolica del Sacro Cuore, 00168 Rome, Italy; 10Biostatistics Unit, IRCCS Regina Elena National Cancer Institute, 00144 Rome, Italy

**Keywords:** IL-8, tumor–stroma interactions, advanced CRC, chemotherapy, anti-angiogenic drugs, prognosis

## Abstract

Among soluble actors that have emerged as druggable factors, the chemokine interleukin-8 (IL-8) has emerged as a possible determinant of response to immunotherapy and targeted treatment in several cancer types; however, its prognostic/predictive role in colorectal cancer (CRC) remains to be established. We: (i) conducted a systematic review of published literature on IL-8 expression in CRC; (ii) searched public transcriptomics databases; (iii) investigated IL-8 expression, by tumor and infiltrating cells, in a series of CRC samples; and (iv) carried out a meta-analysis of published literature correlating IL-8 expression and CRC prognosis. IL-8 possesses an important role as a mediator of the bidirectional crosstalk between tumor/stromal cells. Transcriptomic analysis indicated that specific IL-8 transcripts were significantly overexpressed in CRC compared to normal colon mucosa. Moreover, in our series we observed a statistically significant correlation between PTEN-loss and IL-8 expression by infiltrating mononuclear and tumor cells. In total, 12 papers met our meta-analysis inclusion criteria, demonstrating that high IL-8 levels significantly correlated with shorter overall survival and progression-free survival. Sensitivity analysis demonstrated a highly significant correlation with outcome for circulating, but not for tissue-detected, IL-8. IL-8 is overexpressed in CRC tissues and differentially produced by tumor or stromal components depending on CRC genetic background. Moreover, circulating IL-8 represents a strong prognostic factor in CRC, suggesting its use in the refining of prognostic CRC assessment and potentially the tailoring of therapeutic strategies in individual CRC patients.

## 1. Introduction

Colorectal cancer (CRC) is the third most common malignancy and the second leading cause of cancer death worldwide [1]. In the past 20 years, CRC prognosis has significantly improved due to the preventive/therapeutic application of a much deeper knowledge of the biological mechanisms driving colorectal carcinogenesis and tumor progression [2]. A crucial milestone in systemic CRC treatment is represented by the clinical development of targeted agents interfering with growth factor-driven and neo-angiogenic pathways (such as the epidermal growth factor receptor and the vascular endothelial growth factor [VEGF] and its receptor, respectively) and, more recently, immunotherapy [3,4]. However, resistance to therapy almost inevitably ensues, due to both cancer cell-intrinsic and tumor microenvironment (TME)-mediated mechanisms, and better prognostic/predictive stratification of CRC patients is urgently needed to design therapeutic strategies conveying maximal clinical benefit to individual patients [3,5].

Pro-angiogenic molecules play a relevant role in CRC biology and may significantly impact on its clinical behavior [6]. Interleukin (IL)-8 is a pro-angiogenic, pro-inflammatory C-X-C motif chemokine (CXC) ELR (Glu-Leu-Arg)+, which acts by binding to its cognate receptors (CXC receptor (CXCR) 1 and 2), two G protein-coupled receptors, expressed by both immune/stromal cells and cancer cells [7]. By binding its membrane-bound receptor on the target cells, IL-8 activates specific downstream signaling pathways (such as the phosphoinositide 3-kinase [PI3K] and the mitogen-activated protein kinase [MAPK] cascades), thereby promoting different pro-tumoral phenotypes [8]. Downstream PI3K and MAPK signaling is involved not only in sustaining IL-8 expression, but also in promoting protein translation and cancer cell proliferation and survival. Among the canonical effects caused by tumor-derived IL-8, VEGF-independent angiogenesis is one of most characterized; by binding CXCR1 and CXCR2 expressed by endothelial cells, soluble IL-8 induces endothelial cell proliferation and capillary tube organization [9]. IL-8 also contributes to the epithelial-to-mesenchymal transition (EMT) and cancer stem cells (CSC) generation and maintenance; our group recently reviewed the molecular mechanisms by which IL-8 promotes stemness in the CRC niche [10]. On the other hand, immune cell populations also contribute to IL-8 production, both within the TME and systemically [8]. IL-8 is indeed the major chemoattractant of monocytes/macrophages in the tumor tissue; moreover, tumor-associated macrophages-derived IL-8 promotes the metastatic behavior of CRC cells [11].

Although the biological role of IL-8 in both cancer cell and TME dynamics is well characterized, the clinical impact of IL-8 expression on CRC prognosis remains elusive. A meta-analysis conducted by Xia et al. in 2015 suggested a role for IL-8 expression in CRC diagnosis, staging, and prognosis [12]. Here, we: (i) summarize current knowledge on IL-8’s role in tumor–stroma interactions (TSI); (ii) analyze IL-8 expression in CRC; and (iii) discuss the results of an updated meta-analysis of published studies focusing on the relationship between IL-8 expression levels and outcomes (overall survival [OS] and progression-free survival [PFS]) in advanced CRC.

## 2. Materials and Methods

### 2.1. Analysis of IL-8 Expression Levels in CRC and Normal Samples

Expression data from RNA-sequencing technologies, both for cancer and normal samples, were obtained by interrogating resources made available from large genomic initiatives, such as the Pan-Cancer Analysis of Whole Genomes (PCAWG) and the Genotype-Tissue Expression (GTEx, version4) resources [13,14]. Expression data matrices summarized at the transcript level (Transcript Per Million (TPM) isoform counts) and sample metadata for both datasets (i.e., PCAWG and GTEx) were downloaded from the International Cancer Genome Consortium (ICGC) data portal [https://dcc.icgc.org/releases/PCAWG/transcriptome/; 29 June 2022]. Sample cohorts included in the PCAWG pan-cancer expression data matrix (N = 1359 samples) and the GTEx expression data matrix of normal human tissues (N = 3247 samples) were limited to the following subsets of interest for this study: 51 CRC samples with genomic data labeled as being of optimal quality (histology abbreviation equal to “ColoRect-AdenoCA” and genomic data label equal to “Whitelist” in the PCAWG metadata file) from the PCAWG expression matrix and 88 colon samples from the GTEx expression matrix (histological type equal to “Colon” in the GTEx metadata file) (Appendix A).

Expression levels for the IL-8 gene were extracted from the above-mentioned cohorts of colorectal and normal colon samples based on two transcripts included in the expression data (i.e., ENST00000401931.1, ENST00000307407.3) annotated by Ensembl (www.ensembl.org/ accessed on 1 June 2022) (Appendix A) [15]. For each IL-8 transcript, the two sets of abundance measurements (i.e., the cancer and normal set) were tested for differential expression by using the Wilcoxon rank-sum test, and a test with *p* < 0.05 was considered significant.

### 2.2. Case Selection

CRC samples were obtained from 168 patients who were surgically treated at Regina Elena Cancer Institute between 2000 and 2013. Specifically, 135 colon and 50 rectal carcinomas were retrospectively evaluated and the median follow-up was 50 months. They were staged according to the Unione Internationale Contre le Cancer Tumor-Node-Metastasis (TNM) system criteria [16]. The study was approved by the ethics committee of the Regina Elena National Cancer Institute (10/10/2017, project’s title “Defining the genetic background that modulates angiogenic potential in colorectal cancer models”).

### 2.3. Tissue MicroArray (TMA) Construction and ImmunoHistoChemistry (IHC)

All CRC samples were histopathologically re-evaluated on Haematoxylin and Eosin (H&E) stained slides and representative areas were marked prior to TMA construction. Two core cylinders (1 mm diameter) were taken from selected CRC samples and deposited into two separate recipient paraffin blocks using a specific arraying device (Alphelys, Euroclone, Milan, Italy). In cases where informative results on the TMA were absent due to missing tissue, no tumor tissue, or unsuccessful staining, we reanalyzed the correspondent routine tissue section. Subsequently, 2-µm sections of the resulting microarray block were made and used for IHC analysis after transferring them to SuperFrost Plus slides (Menzel-Gläser, Braunschweig, Germany). IHC staining on the TMA was performed using rabbit monoclonal antibody anti-PTEN (9559, Cell Signaling Technology Inc. Beverly, MA, USA) and rabbit polyclonal antibody anti-IL-8 (Novus Biologicals, Littleton, CO, USA). Immunoreactions were revealed by Bond Polymer Refine Detection on an automated autostainer (Bond™Max, Leica Biosystem, Milan, Italy). Standard processing steps were performed according to the manufacturer’s instructions. PTEN and IL-8 levels in tumor cells were evaluated by a standardized scoring system (0 = no staining or weak cytoplasmic staining in <10% of cells, 1+ = faint/barely perceptible cytoplasmic staining in >10% of cells, 2+ = weak to moderate cytoplasmic staining in >10% of cells, 3+ = strong cytoplasmic staining in >10% of cells); for infiltrating immune cells, IL-8 was assessed in extravascular neutrophils and macrophages using the same scoring system. Evaluation of the IHC results, blinded to all patient data, was performed independently and in a blinded manner by two different investigators (C.A.A. and S.B.). Images were obtained at 20× magnification by using a light microscope (DM2000 LED, Leica) equipped with software able to capture images (Leica).

### 2.4. DNA Extraction

Formalin-fixed, paraffin-embedded (FFPE) tumor specimens were reviewed by a pathologist on H&E stained slides, and main tumor areas were selected and marked based on a minimum tumor cell content of 20%. The corresponding areas of marked tumors were macro-dissected from three to five 5 µm-thick unstained slides.

Paraffin in FFPE sections was removed by Deparaffination Solution (Qiagen, Hilden, Germany). Tissues were then digested by proteinase K at 56 °C overnight and incubated at 90 °C for 5 min to reverse DNA crosslinking. Genomic DNA was extracted on the QIAcube^®^ platform using the QIAamp DNA FFPE tissue kit (Qiagen) according to the manufacturer’s instructions. The extracted DNA was quantified and its quality assessed using NanoDrop^®^ (Thermo Fisher Scientific, Rochester, NY, USA) and Qubit^®^ (Thermo Fisher Scientific) platforms according to the manufacturer’s instructions.

Next generation sequencing (NGS) was performed with a panel (the Ion Ampliseq Cancer Hotspot Panel v2, Thermo Fisher Scientific) composed of 207 amplicons, covering >2800 hotspot mutations in 50 genes: *ABL1*, *AKT1*, *ALK*, *APC*, *ATM*, *BRAF*, *CDH1*, *CDKN2A*, *CSF1R*, *CTNNB1*, *EGFR*, *ERBB2*, *ERBB4*, *EZH2*, *FBXW7*, *FGFR1*, *FGFR2*, *FGFR3*, *FLT3*, *GNA11*, *GNAS*, *GNAQ*, *HNF1A*, *HRAS*, *IDH1*, *JAK2*, *JAK3*, *IDH2*, *KDR*, *KIT*, *KRAS*, *MET*, *MLH1*, *MPL*, *NOTCH1*, *NPM1*, *NRAS*, *PDGFRA*, *PIK3CA*, *PTEN*, *PTPN11*, *RB1*, *RET*, *SMAD4*, *SMARCB1*, *SMO*, *SRC*, *STK11*, *TP53*, and *VHL*. Sequencing was performed on an Ion S5 Sequencer using an Ion 530 Chip and an Ion 530 kit-Chef (all from Thermo Fisher Scientific).

Additional information on data analysis and reporting is available in [17].

### 2.5. Search Strategy

Electronic databases, i.e., PubMed and Embase, were used for conducting the systematic literature search. References reported in the relevant articles, as well as other relevant studies, were also searched manually and scanned for potentially eligible studies. For both the electronic searches, the initial date was January 1st 1994 and the final date was June 30th 2021. The search strategy was designed by three authors (C.B., F.S., and F.C.) and approved by all the other scientists. Boolean operators were used to connect specific keywords. In particular, the specific queries used for the searches were:PubMed: ((((“Interleukin-8”[Mesh]) AND (“Colorectal Neoplasms”[Mesh] AND “Prognosis”[Mesh]) AND (“1994/01/01”[PDat] : “2021/06/30”[PDat]))) OR (((((Prognosis) AND ((Colorectal Neoplas* OR CRC OR Colorectal Cancer OR Colorectal tumor)))) AND (((Interleukin-8 OR IL-8 OR interleukin8 OR IL8)))) AND (“1994/01/01”[PDat] : “2021/06/30”[PDat])))(‘colorectal neoplasm’/exp OR ‘colorectal neoplasm’ OR (colorectal AND (‘neoplasm’/exp OR neoplasm)) OR crc OR ‘colorectal cancer’/exp OR ‘colorectal cancer’ OR (colorectal AND (‘cancer’/exp OR cancer)) OR ‘colorectal tumor’/exp OR ‘colorectal tumor’ OR (colorectal AND (‘tumor’/exp OR tumor))) AND (‘interleukin 8’/exp OR ‘interleukin 8’ OR ((‘interleukin’/exp OR interleukin) AND 8) OR ‘il 8’/exp OR ‘il 8’ OR (il AND 8) OR interleukin8 OR il8) AND (1994:py OR 1995:py OR 1996:py OR 1997:py OR 1998:py OR 1999:py OR 2000:py OR 2001:py OR 2002:py OR 2003:py OR 2004:py OR 2005:py OR 2006:py OR 2007:py OR 2008:py OR 2009:py OR 2010:py OR 2011:py OR 2012:py OR 2013:py OR 2014:py OR 2015:py OR 2016:py OR 2017:py OR 2018:py OR 2019:py OR 2020:py OR 2021:py) AND (‘bevacizumab’/dd OR ‘monoclonal antibody’/dd) AND ‘colorectal cancer’/dm AND (‘clinical study’/de OR ‘clinical trial’/de OR ‘cohort analysis’/de OR ‘controlled clinical trial’/de OR ‘controlled study’/de OR ‘meta analysis’/de OR ‘multicenter study’/de OR ‘phase 2 clinical trial’/de OR ‘phase 2 clinical trial topic’/de OR ‘phase 3 clinical trial’/de OR ‘phase 3 clinical trial topic’/de OR ‘prospective study’/de OR ‘randomized controlled trial’/de OR ‘randomized controlled trial topic’/de OR ‘retrospective study’/de) AND ‘avastin’/tn

### 2.6. Inclusion and Exclusion Criteria

The inclusion criteria were as follows:Original papers;All IL-8 expression evaluations, including serum, plasma, and tissue;Studies with hazard ratio (HR) and 95% confidence intervals (95% CI) for OS or PFS.

The exclusion criteria were as follows:Papers reporting data which cannot be extracted;Non-English papers;Non-full-length papers;Reviews, systematic reviews, meta-analysis, book chapters, congress abstracts;Animal studies or basic research papers;Studies on gene polymorphisms;Studies also containing other histotypes different from CRC;Papers not relevant to the topic;Note publications, retracted articles.

Out of 436 studies retrieved by Pubmed, Embase, and other searches, a total of 12 studies correlating IL-8 expression with survival outcomes (either OS or PFS) in CRC patient series were found to match our selection criteria. Reasons for exclusion are detailed in the PRISMA flowchart reported in Appendix A.

Data collection was performed by C.B., M.M., and F.C. and included: title, first author, year, method of IL-8 detection, number of patients (total, low, and high IL-8), HR, 95% CI, *p*-value, clinical endpoints, and treatments. The principal methods used for assessing IL-8 expression were specific ELISA assays or multiplex cytokine arrays for plasma/serum samples and protein lysate arrays for tissue samples.

## 3. Results

### 3.1. IL-8 in TSI

We conducted a systematic review of preclinical data on the role of IL-8 production in the complex scenario of TSI (Figure 1). IL-8 production is regulated at three main levels: (i) repression/activation of a gene promoter, (ii) mRNA stabilization, and (iii) post-translational cleavage of its precursor [17,18,19]. Many groups, including ours, have shown that IL-8 production by cancer cells depends on their genetic/molecular background. Ackermann and colleagues showed that the knock-down of spectrin alpha (non-erythrocytic 1 (SPTAN1), a cytoskeletal protein involved in DNA repair) results in increased IL-8 production by CRC cells in vitro. SPTAN1-dependent IL-8 production, in turn, increases neutrophils’ migration into the TME (Figure 1A) [20]. Conversely, Wu et al. demonstrated that leukemia inhibitory factor receptor (LIFR) silencing downregulates IL-8 levels; IL-8 silencing, in turn, reduced LIFR-dependent angiogenesis, thus indicating a bidirectional crosstalk between IL-8 and LIFR (Figure 1A) [21]. Another line of evidence establishes a link between IL-8 and the tumor suppressor *SMAD4*, a downstream mediator of TGF-β that is mutated in 5–20% of CRC patients (loss of heterozygosity) [22]. Ogawa and coworkers recently showed that transient SMAD4 knock-down results in IL-8 overexpression in CRC cells. Consistently, IL-8 was significantly decreased by enforced expression of SMAD4 in SMAD4-deficient cell lines [23]. In this context, IL-8 induction by SMAD4 knock-down results in neutrophil recruitment into the TME via the IL-8 receptor CXCR2 (Figure 1A); these tumor-associated neutrophils (TAN), in turn, overexpress IL-8 when compared to peripheral blood neutrophils [23]. Finally, our group recently demonstrated that *BRAF* mutations and PTEN-loss increase IL-8 production in preclinical models of CRC through CHOP-dependent transcriptional upregulation (Figure 1A) [17].

Tumor-derived IL-8 displays autocrine activities, sustaining CRC tumorigenesis and treatment resistance. IL-8 maintains and promotes cell stemness through Snail-dependent EMT, among other mechanisms (Figure 1B) [10]. On the other hand, Oct4, a specific marker of CRC stem cells, promotes IL-8 upregulation [24]. With regard to its autocrine activities, IL-8 also plays a role in chemoresistance. Doxorubicin-resistant CRC cells, compared to doxorubicin-sensitive ones, display higher levels of IL-8, and IL-8 silencing restores chemosensitivity by downregulating the expression of multidrug resistance 1 (Figure 1C) [25]. Other studies demonstrated that, in addition to favoring tumor progression, IL-8 is also involved in resistance to oxaliplatin [26].

IL-8 is also produced by stromal/immune cells, both in the TME and systemically, under stimulation by cancer cells. Watanabe and coworkers showed that cancer associated-fibroblasts (CAF) secrete higher IL-8 levels compared to normal fibroblasts or CRC cell lines. In particular, the addition of recombinant Chitinase 3-like 1 (CHI3L1), a protein that promotes cell proliferation and angiogenesis, correlates with the increased IL-8 levels released by CAF (Figure 1D) [27]. In early CRC, production of both IL-8 and IL-6 by myofibroblasts in the TME is responsible for cancer stem cells expansion and maintenance through the regulation of Notch-STAT3 pathways (Figure 1B) [28]. Amorim and colleagues recently showed that extracellular vesicles released by breast tumor cells are associated with TAN polarization towards a pro-tumoral N2 phenotype, characterized by the expression of CD184, IL-8, VEGF, and MMP-9, thereby sustaining chronic inflammation (Figure 1E) [29]. Similarly, exosomes derived by *KRAS*-mut CRC cells induce IL-8 upregulation and neutrophil enrichment [30].

Accumulating evidence clearly indicates that IL-8 levels bear important clinical implications in CRC. Several groups have reported that CRC patients display higher levels of IL-8 compared to healthy subjects, and that IL-8 levels are negative prognostic/predictive factors for chemotherapy [31]. Furthermore, high serum IL-8 levels are associated with the expression of specific CD4+ T cell genes in CRC patients (Figure 1F) [32]. Oladipo et al. showed that the majority of CRC tumor cores (65.4%) expressed IL-8 within the tumor-associated inflammatory infiltrate, whereas virtually none of the normal colorectal tissue cores showed detectable IL-8 expressing inflammatory cells [33]. IL-8 positivity in tumor infiltrate was also independently associated with a lower risk of disease recurrence (HR for relapse-free survival (RFS): 0.55, CI: 0.36–0.85, *p* < 0.001); median RFS was indeed not reached in stage II-III CRC patients whose tumors displayed an IL-8 positive infiltrate, as compared to a median RFS of 69 months in patients with no IL-8 immunoreactivity in their tumor infiltrate (*p* < 0.01 by log-rank test).

### 3.2. IL-8 Expression in CRC Patients

#### 3.2.1. IL-8 Expression Analysis in RNA-Sequencing Data Resources

To gain further insights on IL-8 expression in CRC, we interrogated public atlases of gene expression that have been obtained by RNA-sequencing technology and which have been made available from large genomic initiatives for both cancer and normal human tissues, such as the PCAWG and the GTEx resources [13,14]. Expression values were evaluated at the transcript level for two transcripts annotated for the *IL-8* gene which were available in the interrogated cohorts of RNA-sequencing data (i.e., ENST00000401931.1 and ENST00000307407.3). Both IL-8 transcripts show significant increases in expression levels (Wilcoxon rank-sum test *p* = 3.4 × 10^−13^ and *p* = 4.6 × 10^−19^ for IL-8 transcript isoforms ENST00000401931.1 and ENST00000307407.3, respectively) in CRC compared to normal colon tissues (Figure 2A and Table 1).

#### 3.2.2. IL-8 Analysis in Tumor Specimens from CRC Patients

We explored correlations between *BRAF*/PTEN status and IL-8 expression (scored separately in tumor cells and infiltrating mononuclear cells) by IHC in a series of 168 CRC patients; bio-pathological patient characteristics are summarized in Appendix A. Eleven (7%) and sixty (36%) patients were found to have mutated/altered *BRAF* and PTEN, respectively.

With the limits of a small number of *BRAF* mutant cases in our series (N = 11), no statistically significant correlation was found between *BRAF* mutations and IL-8 expression by either tumor or infiltrating mononuclear cells. However, when we analyzed IL-8 expression in both tumor and infiltrating cells (i.e., high (H)/low (L), H/H, L/L, L/H, respectively), we observed a statistically significant correlation with PTEN status (*p* = 0.02) (Figure 2B,C). This association is linked to a strong correlation that was found between PTEN-loss and a lack of IL-8 in infiltrating mononuclear cells (*p* = 0.003) (Appendix A). On the contrary, PTEN status was not significantly associated with IL-8 expression by tumor cells (*p* = 0.72) (Appendix A).

### 3.3. Meta-Analysis: IL-8 and CRC Prognosis

The selection of papers included in the meta-analysis is detailed in the PRISMA flowchart (Appendix A). Overall, OS data were available for a total of 1967 patients, while PFS data were available for a total of 1555 patients. Details on the 12 studies analyzed are reported in Appendix A: 11/12 studies selected reported on metastatic CRC patients who received systemic chemotherapy with (7 studies) or without (3 studies) an anti-angiogenic agent (bevacizumab in 6 studies, cediranib in 1 study) or anti-angiogenic treatment without chemotherapy (regorafenib in 2 study); 1 study reported on primary CRC, undergoing curative surgery and receiving adjuvant chemotherapy according to standard indications [34]; 10 studies concluded that there was a variable degree of association between IL-8 expression levels and worse outcomes. Using the Newcastle–Ottawa quality assessment scale, computed separately by four different authors (C.B., M.M., I.S., F.C.), two studies were deemed to be of high quality (score > 7), eight to be of moderate quality (score > 3 and <7), and two to be of low (score 3) quality (Appendix A).

#### 3.3.1. Primary Analysis: OS

Pooled analysis of unadjusted OS data (seven studies, 1227 patients) revealed a significantly higher risk of death for CRC patients with high *versus* low IL-8 expression (pooled HR by random effect model: 2.804; 95% CI: 1.675–4.694; *p* < 0.001) (Figure 3A). These results were confirmed in the seven studies (1066 patients) reporting adjusted OS data (pooled HR by random effect model: 2.025; 95% CI: 1.289–3.180; *p* = 0.002) (Figure 3B). In both cases (unadjusted and adjusted), significant heterogeneity between studies was detected (I2: 86.058 and 71.909, respectively). Overall (14 studies, 2293 patients), despite significant heterogeneity between studies (I2: 81.612), high IL-8 expression significantly correlated with worse OS in CRC patients (pooled HR by random effect model: 2.376; 95% CI 1.690–3.341; *p* < 0.001) (Figure 3C). The funnel plot was visually asymmetrical towards positive associations, suggesting the presence of small-study effect (Appendix A).

#### 3.3.2. Secondary Analysis: PFS

Pooled analysis of the eight studies (1710 patients) reporting PFS data revealed a significantly higher risk of progression or death for CRC patients with high versus low IL-8 expression (pooled HR by random effect model: 1.639; 95% CI: 1.199–2.239; *p* = 0.002) (Figure 4A). The association between high IL-8 expression and worse PFS was confirmed by analyzing only the three studies reporting adjusted PFS data (675 patients, Figure 4B), although it did not reach statistical significance using a random effect model. In both cases, significant heterogeneity between studies was detected (I2: 80.392 and 85.456, respectively). As for OS, the funnel plot was visually asymmetrical towards positive associations, suggesting the presence of small-study effect (Appendix A).

#### 3.3.3. Exploratory Analysis: Sensitivity Analysis According to IL-8 Source and the Type of Systemic Treatment

Circulating levels of IL-8 in plasma or serum were assessed in 11 studies; high IL-8 expression significantly correlated with worse OS (12 studies, 1983 patients; pooled HR by random effect model: 2.841; 95% CI: 2.376–3.398; *p* < 0.001; Figure 5A) and PFS (6 studies, 1400 patients; pooled HR by random effect model: 1.982; 95% CI: 1.525–2.575; *p* < 0.001; Figure 5B). Heterogeneity between studies was moderate for OS (I2: 14.198) and high for PFS (I2: 56.656). Conversely, in the single study assessing tissue expression levels of IL-8 (155 patients), no significant impact on either OS or PFS was detected (Figure 6D,E) [35]. Five studies (466 patients) analyzed the impact of IL-8 expression in CRC patients who had all received treatment with anti-angiogenic agents (with or without chemotherapy); high IL-8 expression significantly correlated with worse OS (pooled HR by random effect model: 3.427; 95% CI: 2.446–4.802; *p* < 0.001; Figure 6A) and PFS (pooled HR by random effect model: 3.237; 95% CI: 1.684–6.219; *p* < 0.001; Figure 6B), with low heterogeneity between studies for OS and high heterogeneity for PFS (I2: 1.718 and 55.180, respectively). Similarly, a significant negative impact of IL-8 expression on OS was observed in the five studies (735 patients) in which CRC patients received chemotherapy without anti-angiogenic agents (pooled HR by random effect model: 2.185; 95% CI: 1.571–3.039; *p* < 0.001; Figure 6C), without heterogeneity (I2: 0.000). No data on PFS were available in these studies. No significant effect of IL-8 expression on either OS or PFS was observed in the two series (783 patients) that included CRC patients treated both with and without anti-angiogenic agents (Figure 7A,B) [35,36].

## 4. Discussion

In this manuscript, we evaluated the multifaceted role played by IL-8 in TSI and the clinical implications in terms of CRC prognosis and response to therapy. Data presented here confirm a relationship between high IL-8 expression levels and worse outcome (both PFS and OS) in CRC, particularly in advanced disease undergoing systemic therapy. The exploratory analyses conducted suggest that circulating, rather than tissue-detected, IL-8 levels have a solid relationship with outcome and bear a prognostic, rather than predictive, significance. Indeed, such a relationship appears to be independent from the type of systemic treatment applied (chemotherapy ± anti-angiogenic agents). Nevertheless, the evaluation of IL-8 source in CRC tissues could help to better characterize TME compositions and functions in terms of tumor progression and drug response.

We recently demonstrated that IL-8 production by cancer cells in vitro is tightly regulated at multiple levels and is strongly dependent on the genetic landscape (i.e., *BRAF* and PTEN status) [17]. However, we could not confirm the association between *BRAF* mutations and IL-8 production in a series of surgical CRC samples, most likely due to the low number of *BRAF* mutant cases. Conversely, we interestingly demonstrated that PTEN-loss is associated with statistically significant variations in IL-8 expression, particularly by mononuclear tumor-infiltrating cells (Figure 2). Indeed, cancer cells are not the only source of IL-8 secretion (Figure 1). Thus, cellular composition of the TME (i.e., stromal and immune cells) and the ability of its components to release IL-8 may be of crucial importance for understanding its biological and clinical significance (Figure 1). Since IL-8 demonstrates pleiotropic effects in TSI, the importance of precisely defining the source of IL-8, and possibly correlating its levels with clinical parameters, has increasingly emerged over the years. Advanced technologies applied to different cancer histotypes has allowed for the direct correlation of soluble mediators with their specific source in the tissue. Furthermore, single-cell RNA-sequencing demonstrated a difference in therapeutic response according to the source of IL-8; IL-8 was produced by myeloid and lymphoid cells in patients not responding to immune checkpoint inhibitors (ICI) but not in those treated with chemotherapy or anti-angiogenic agents and was significantly correlated with worse OS in patients affected by urothelial cancer and renal cell carcinomas [28]. In this context, targeting of IL-8 and its cognate receptors might be developed into a potentially effective treatment strategy, as suggested, for example, by a recent phase I trial with an anti-IL-8 mAb which, in monotherapy, achieved disease control (for 3.5, 4.5, and 7 months) even with 3/4 advanced, refractory CRC patients [37]. Drugs targeting the IL-8 receptors CXCR1 and CXCR2 are also in clinical development; for example, two different clinical trials recruiting HER-2 negative metastatic breast cancer are currently evaluating the role of the CXCR1/CXCR2 inhibitor reparixin in combination with chemotherapeutic paclitaxel (NCT02370238; NCT02001974).

A previous meta-analysis published by Xia et al. in 2015 concluded that high IL-8 levels significantly correlated with advanced CRC stage and increased mortality risk when compared to low levels (N = 1215; pooled HR by random effect model 1.54, 95% CI 1.03–2.32) [12]. In a subgroup of four high-quality studies specifically dealing with stage IV disease (N = 221), the pooled HR for death was 2.28 (95% CI 1.60–3.25), with no significant heterogeneity apparent. Four of the fifteen studies included in this previous meta-analysis matched our selection criteria and were included in the present meta-analysis [35,38,39,40]. We also identified eight additional studies, six of which were published after August 17th, 2014 (the date of the last search for the previous meta-analysis). Even in a larger population of patients (1967 for OS and 1555 for PFS) and with a different composition of contributing studies, HR for OS in our study (2.376; 95% CI 1.690–3.341) was strikingly consistent with the one reported in the study by Xia for metastatic patients, supporting the rather robust prognostic role of IL-8 in CRC, independent of the methodology used to analyze the published literature. Thus, the results of our study support the idea that circulating IL-8, detected in plasma or serum, is more relevant to its prognostic significance. Such results raise the issue of standardization regarding the source and methods of IL-8 detection, along with times of sample collection (at diagnosis, pre- and post-treatment, etc.) and patient characteristics (stage, type of treatment, etc.), in order to collect homogeneous clinical data. This point may be biologically, in addition to methodologically, relevant, as the source of IL-8 production might reveal itself to be crucially important. The relationship between high IL-8 expression and worse outcomes for advanced CRC was qualitatively similar in cohorts of patients receiving treatment with chemotherapy + anti-angiogenic agents (i.e., bevacizumab, cediranib, rogorafenib [38,41,42,43,44]), chemotherapy alone ([39,40,45]), or anti-angiogenic agents alone [44], suggesting a prognostic rather than predictive effect. Two studies assessed IL-8 prognostic performance in mixed populations of patients treated with chemotherapy ± anti-angiogenic agents. In the study by Bruhn et al., IL-8 levels were neither prognostic nor predictive of the benefit of bevacizumab, but the fact that IL-8 was measured in tissue lysates leaves open the possibility (discussed above) that the source of IL-8 might have affected the results [35]. In the study by Spencer et al., circulating IL-8 levels were prognostic for both PFS and OS in the overall population, but not predictive of benefit from the addition of cediranib to standard chemotherapy [36]. Similar results were observed in the study by Tabernero et al., in which the benefit of regorafenib *versus* placebo was similar among high and low circulating IL-8 levels, regardless of the cut-off used [46]. Interestingly, in the same study, IL-8 levels were significantly associated with both PFS and OS in the placebo group. Overall, the evidence presented here strongly indicates that IL-8 expression in plasma/serum conveys prognostic, rather than predictive, information in advanced CRC. This conclusion is further supported by the prognostic role of circulating IL-8 in surgically operated, early-stage CRC patients, in which elevated IL-8 levels were independently associated with worse PFS (HR 1.82, 95% CI 1.19–2.76) and OS (HR 2.33, 95% CI 1.32–4.11) [34]. Moreover, in the study by Xiao et al. (not included in the present meta-analysis due to failure to correctly calculate 95% CI for OS), high IL-8 expression was independently associated with disease-free survival and OS through multivariate analysis in another series of surgically resected CRC patients [47]. Furthermore, the authors also confirmed such prognostic value in silico, using the GSE33113 dataset from The Cancer Genome Atlas project [13].

The prognostic role of IL-8 has also been hypothesized in other tumor types treated with an array of different therapeutic strategies. In a study from our group, IL-8 was the circulating factor most significantly correlated with survival in both a discovery and an independent validation cohort of advanced pancreatic cancer treated with nanoliposomal irinotecan-based II-line chemotherapy [48]. A large retrospective study showed that elevated baseline IL-8 serum levels significantly correlated with worse OS with both ICI (i.e., atezolizumab) and chemotherapy in metastatic urothelial cancer, as well as with atezolizumab monotherapy, for patients with metastatic renal cell carcinomas; there was no correlation with combined atezolizumab/bevacizumab tyrosine kinase inhibitors [28]. Similarly, Schalper et al. demonstrated a strong negative correlation between high serum IL-8 and OS in a large cohort of patients from four clinical trials of advanced renal cell carcinoma, melanoma, and non-small cell lung cancer treated with ICI, chemotherapy, or mTOR inhibitors, regardless of the applied treatment [49]. These findings clearly suggest that IL-8 could be a crucial prognostic factor in a wide variety of tumors of different histological origin where treatment with chemotherapy, targeted therapy, or immunotherapy is currently undertaken.

## 5. Conclusions

Overall, the results presented here clearly suggest two main aspects regarding IL-8 evaluation in CRC biology: (i) an activated IL-8 network in the TME of CRC tissues profoundly modifies cancer and non-cancer cell behavior; thus, identifying the source of IL-8 production is necessary to define TSI, which in turn may impact on tumor progression and, eventually, may inform the development of specific IL-8 axis-directed therapeutic strategies. (ii) routine assessment of circulating IL-8 levels could be implemented in CRC in order to stratify patients with different prognoses and select the most appropriate treatment strategy.

## Figures and Tables

**Figure 1 biomedicines-10-02631-f001:**
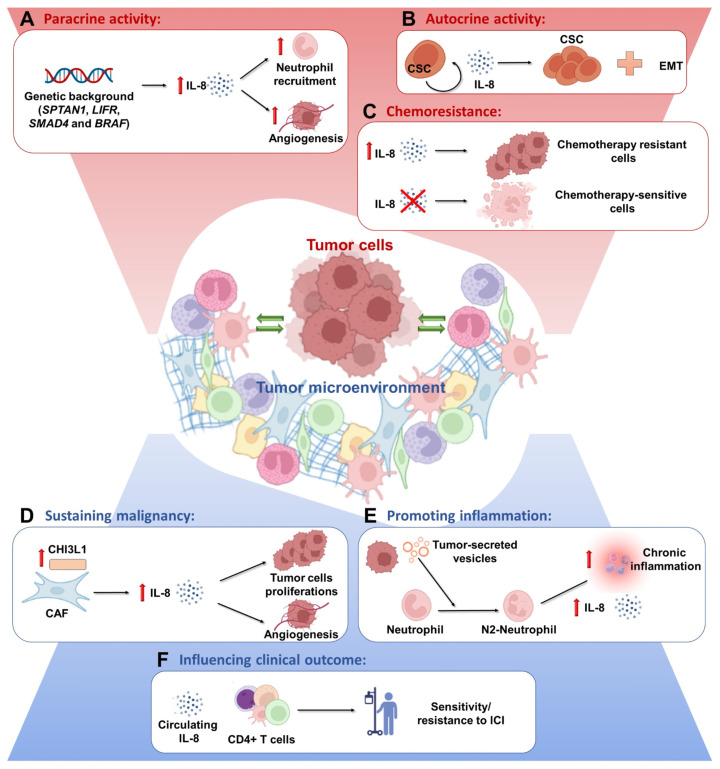
**Functional consequences of IL-8-mediated crosstalk between tumor and stromal/immune cells within the TME.** In the TME, both tumor and stromal/immune cells can release IL-8, which in turn acts as an autocrine or paracrine soluble factor. By mediating TSI, the IL-8 axis affects drug resistance, tumor progression, and clinical outcome in CRC patients. **Top panel (in red):** IL-8 production by CRC cells during tumor progression is influenced by their genetic background and regulated by the activity of genes such as *SPTAN1*, *LIFR*, *SMAD4*, or *BRAF*. Once released in the TME, tumor cells-derived IL-8 sustains the acquisition of a local pro-tumoral niche by stimulating neutrophil recruitment and promoting angiogenesis in a paracrine fashion (**A**). At the same time, IL-8 sustains CSC proliferation and EMT in an autocrine fashion (**B**), resulting in increased tumor aggressiveness and chemoresistance (**C**). **Bottom panel (in blue):** On the other hand, IL-8 can be produced by TME components and modulate TSI, inflammation, and immune response. For example, CAF may stimulate IL-8 release, thereby sustaining CSC proliferation and further promoting angiogenesis locally (**D**); under stimulation by tumor-secreted micro-vesicles, pro-tumoral N2-neutrophils promote chronic inflammation (**E**) both locally and systemically, resulting in high levels of IL-8 in the circulation and tissues which, in turn, influences the composition of tumor-infiltrating immune populations and sensitivity/resistance to ICI (**F**).

**Figure 2 biomedicines-10-02631-f002:**
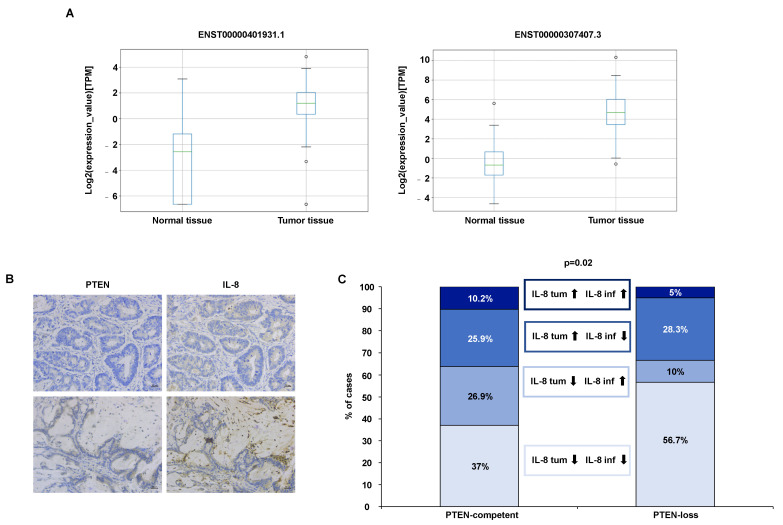
**Analysis of IL-8 in CRC patients.** (**A**): IL-8 expression in normal and CRC tissues. Boxplots show expression values of the IL-8 gene across 51 CRC samples from the PCAWG resource (“tumor”) and 88 colon samples from the GTEx resource (“normal”) for two transcripts annotated for this gene: ENST00000401931.1 (left panel) and ENST00000307407.3 (right panel). Expression values are given as log2-transformed transcript per million (TPM). (**B**): Representative CRC cases showing PTEN-loss/IL-8 infiltrate low and IL-8 tumor high images (top panels) and PTEN competent/IL-8 infiltrate high and IL-8 tumor images (bottom panels). Scale bar = 30 μm. Patients carrying *PTEN* gene inactivating alterations or completely lacking PTEN protein expression (IHC score 0) are referred to as PTEN-loss. (**C**): Distribution of PTEN competent and PTEN-loss patients according to IL-8 tumor and infiltrate levels. Arrows indicate low or high levels of IL-8.

**Figure 3 biomedicines-10-02631-f003:**
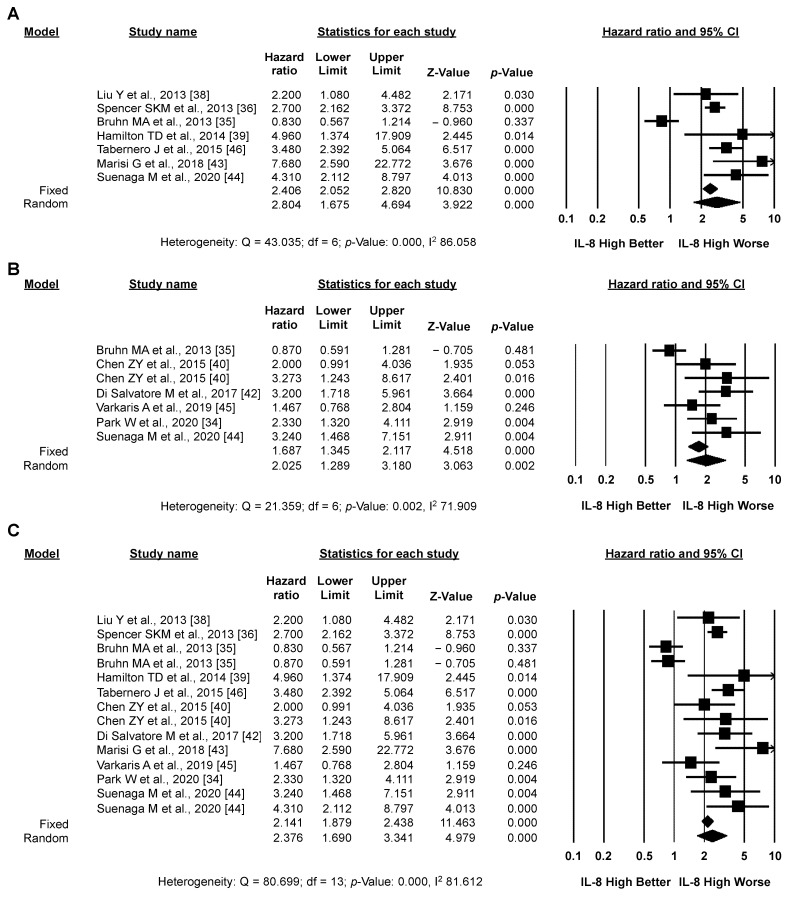
**High IL-8 levels in the circulation and tissues correlates with worse OS.** Meta-analysis of the HR for the association of high levels of IL-8 and unadjusted (**A**), adjusted (**B**), or unadjusted/adjusted (**C**) OS data.

**Figure 4 biomedicines-10-02631-f004:**
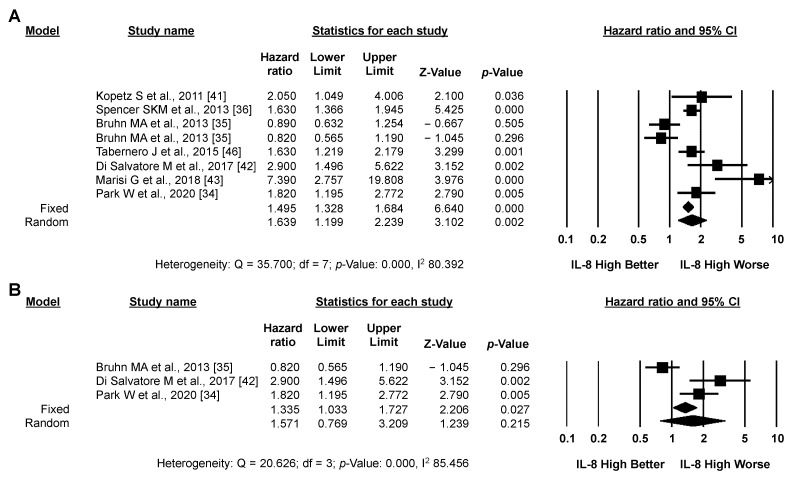
**High IL-8 levels in the circulation and tissues correlates with worse PFS.** Meta-analysis of the HR for the association of high levels of IL-8 and unadjusted/adjusted (**A**) or adjusted (**B**) PFS data.

**Figure 5 biomedicines-10-02631-f005:**
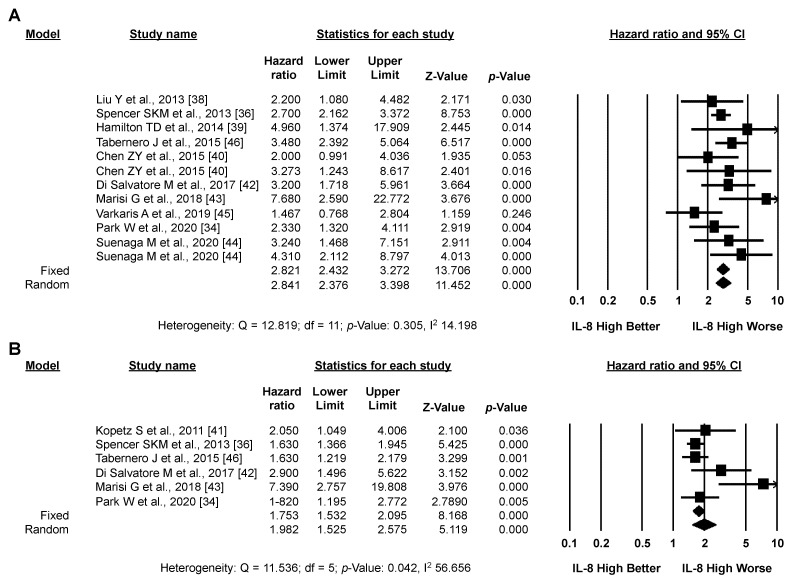
**High IL-8 levels in the circulation correlates with worse OS and PFS.** Meta-analysis of the HR for the association of the source of IL-8 (plasma or serum) and OS (**A**) or PFS (**B**) data.

**Figure 6 biomedicines-10-02631-f006:**
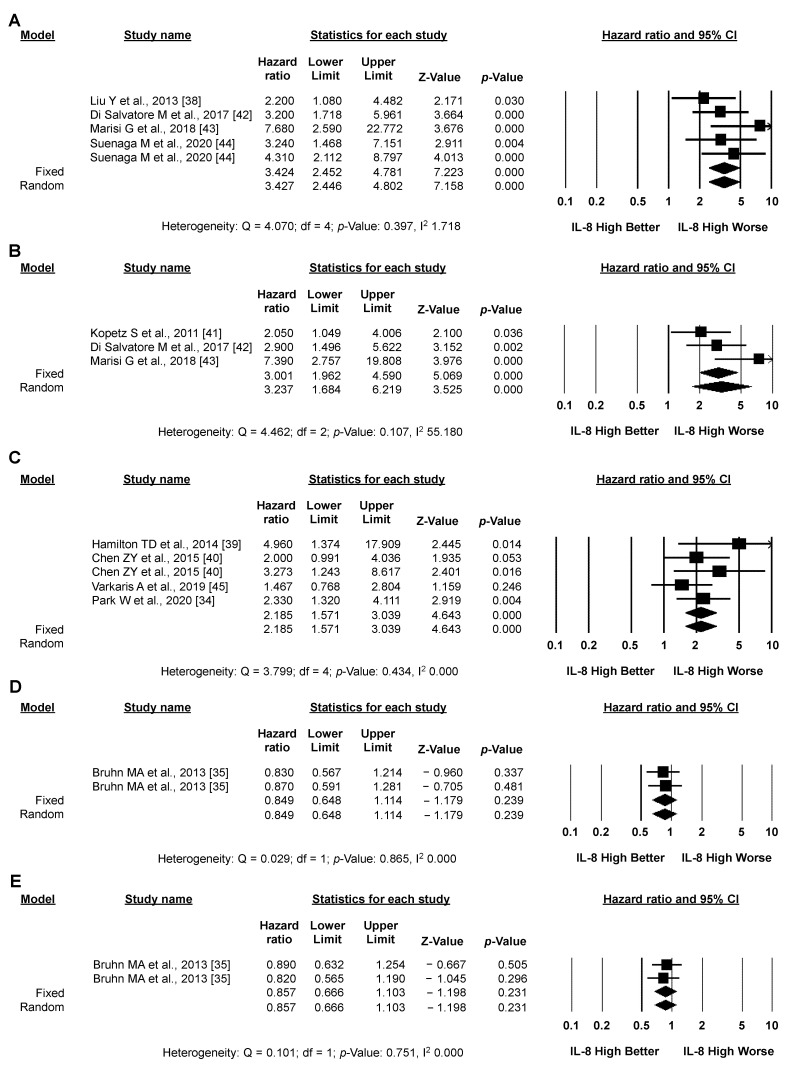
**High IL-8 levels in the circulation, but not tissues, correlates with the clinical outcome of systemic treatment.** Meta-analysis of the HR for the association of circulating IL-8 in relation to OS (**A**,**C**) or PFS (**B**) and patients who received anti-angiogenic agents (**A**,**B**) or chemotherapy without anti-angiogenic agents (**C**). Meta-analysis of the HR for the association of IL-8 found in tissues and OS (**D**) or PFS (**E**) data.

**Figure 7 biomedicines-10-02631-f007:**
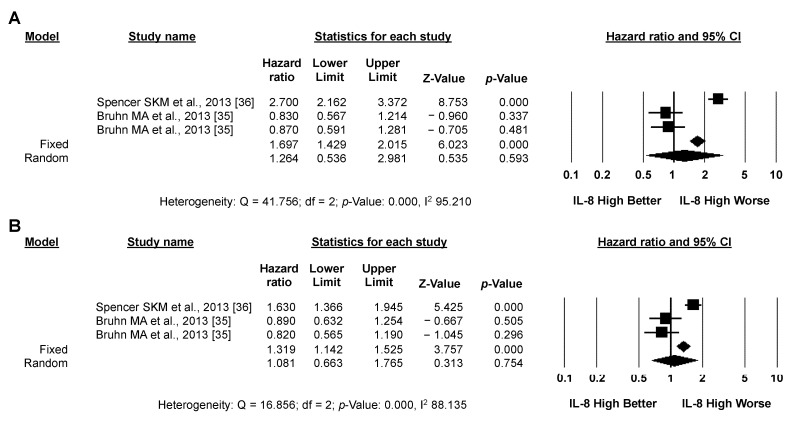
**IL-8 levels do not correlate with the clinical outcome of systemic treatment with/without anti-angiogenic agents.** Meta-analysis of the HR for the association of the OS (**A**) or PFS (**B**) source and patients who had received chemotherapy with or without anti-angiogenic agents.

**Table 1 biomedicines-10-02631-t001:** IL-8 mean expression values in normal colon tissues and CRC samples.

Gene	Transcript (Ensembl Identifier)	Mean_Normal (TPM)	Mean_Tumor (TPM)
IL-8 (ENSG00000169429)	ENST00000307407.3	1.9	80.6
IL-8 (ENSG00000169429)	ENST00000401931.1	0.4	3.6

The table lists two transcripts for the *IL-8* gene (Ensembl gene identifier: ENSG00000169429). Expression values were obtained from the GTEx and PCAWG resources for normal colon tissues and CRC samples, respectively. Full sets of transcript abundance for the IL-8 isoforms are available in Appendix A. TPM, transcript per million.

## Data Availability

All data supporting the findings of this study are available in the article along with Appendix A. Additional information will be kindly provided without any restrictions.

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
