# Peer review of "Interleukin-8 in Colorectal Cancer: A Systematic Review and Meta-Analysis of Its Potential Role as a Prognostic Biomarker"

_biomedicines, 2022, doi:10.3390/biomedicines10102631_

Round 1

Reviewer 1 Report

The presented manuscript provides important information on the role of IL-8 in different clinical scenarios. The manuscript is well written and provides insight into the complex roles played by IL-8 in colorectal cancer.

The only major point of contention is Figure 1 which attempts to show different cellular and molecular scenarios in which IL-8 would be involved. While it is a figure that could recapitulate the different molecular functions of IL-8 it is not clearly explained. The figure caption is extremely terse and in the body of the manuscript there is not enough information about this figure. It is recommended that the authors write a figure caption with sufficient information to fully explain the different molecular functions of IL-8.

Author Response

Dear Editor,

Enclosed please find a thoroughly revised version of the manuscript entitled “Interleukin-8 in colorectal cancer: A systematic review and meta-analysis of its potential role as a prognostic biomarker”, that we wish to resubmit for publication in Biomedicines.

First, we would like to thank the Reviewers for their helpful and constructive comments. We have addressed Reviewer's comments and believe that the revised manuscript has substantially improved. We thus hope it can now be published in your prestigious Journal.

Reviewers #2 and #4 have raised some editorial comments that we wish to address first and on which we would like to seek Editor's advice:

1) Reviewer #2 highlights this point of the Supplementary Figures:

[...] many results are presented in supplement which makes it even harder to get the overview.

In the revised version, we have moved two of the Supplementary Figures into the main manuscript. Nevertheless, if the Editor/Editorial Office deems it appropriate, we can consider to move all the Supplementary Figures into the main manuscript, in order to help readers interpreting the data.

2) Reviewer #4 highlights the two following points:

In reference section please keep the journal writing style (as per Biomedicines) CONSISTENT for all the listed references

We added the reference by using Endnote, according to MDPI style. However, if the Editorial Office deems it appropriate, we can modify the reference style according to the instructions you will provide for the Journal's preferred style.  

Text size throughout the review is not consistent, please correct accordingly

We formatted the manuscript according to the layout available at: https://www.mdpi.com/journal/biomedicines/instructions; however, if the Editorial Office deems it appropriate, we can modify manuscript's formatting according to the instructions you will provide for the Journal's preferred style.  

A point-by-point reply to Reviewers' comments is attached below for your convenience.

REVIEWER #1:

The presented manuscript provides important information on the role of IL-8 in different clinical scenarios. The manuscript is well written and provides insight into the complex roles played by IL-8 in colorectal cancer.

We thank the Reviewer for his/her recognition of the relevance of our report.

The only major point of contention is Figure 1 which attempts to show different cellular and molecular scenarios in which IL-8 would be involved. While it is a figure that could recapitulate the different molecular functions of IL-8 it is not clearly explained. The figure caption is extremely terse and in the body of the manuscript there is not enough information about this figure. It is recommended that the authors write a figure caption with sufficient information to fully explain the different molecular functions of IL-8.

We thank the Reviewer for raising this important point. We have extensively revised Figure 1, adding lettered panels to increase readability and streamline the information flow; this is reflected in a more descriptive revised Figure Legend, as well as in the revised manuscript text (revised paragraph 3.1, lines 258-320), which now refers to the different figure panels specifically.

REVIEWER #2:

The study by Bazzichetto et al, “Interleukin-8 in colorectal cancer: A systematic review and meta-analysis of its potential role as a prognostic biomarker” is a combination of a) literature review 2) meta-analysis c) database transcriptome analysis 4) CRC sample analysis

I have the following comments:

  • The combination of 1-4 above give a confusing impression, is complex and makes it very hard to penetrate and review the manuscript.
  • I can’t see that this paper comes up with any other results than those published by Xia et al 2015, as mentioned on line 69-70, “A meta-analysis conducted by Xia et al. in 2015 suggested a role for IL-8 expression in CRC diagnosis, staging, and prognosis”.
  • I suggest that the authors separate the literature study from their own original data and try to write two papers instead.

We thank the Reviewer for emphasizing these critical points, which we ourselves took into consideration when planning the manuscript. The meta-analysis by Xia et al. did suggest that IL-8 conveys a negative prognostic impact in CRC, but was never validated by independent groups; moreover, it dates back to almost 7 years ago: indeed, we considered at least 6 additional papers published after the meta-analysis by Xia et al. and excluded some of the papers included in their work, because they did not match our selection criteria. Even though conducted on a different pool of patients' data, our results confirm the negative prognostic role of IL-8 (thus substantially strengthening previous findings) and provide sensitivity analyses suggesting, for the first time, a role for IL-8 source (tissue vs. circulating) in patients treated with either chemotherapy alone or chemotherapy plus an anti-angiogenic agent. Overall, we believe that such information adds to current knowledge on the topic and deserves being reported. This conclusion is supported by the fact that, even though information on the IL-8 prognostic impact in CRC has been available for quite some time, this has not led to the adoption of circulating IL-8 measurement (easily available and relatively inexpensive) in the routine clinical care of CRC patients. This indeed prompted us not only to strengthen previous results by performing an updated meta-analysis, but also to integrate multiple types of data (i.e. transcriptomic analysis, original patient series, and published evidence) in order to convey a more comprehensive idea of the importance of IL-8 in CRC biological and clinical behavior. Although we agree with the Reviewer that such approach makes reading and interpreting the information contained in the manuscript more challenging, we are also convinced that finding all relevant information in one place would give readers a clearer perception of the potential role of IL-8 in CRC and help clinicians and researchers alike design new clinical trials and translational experiments, including standardized IL-8 assessment as a new relevant tool in CRC management. 

The Introduction is lacking a background about the cytokine IL-8 in normal and CRC cells: function, role and cell types producing IL-8 (both intracellular and soluble), cell types expressing receptors and the effect of IL-8 ligation through CXCR1/2. The introduction is also lacking the role of the different signalling mechanisms involved in CXCR1/2 pathways.

We thank the Reviewer for raising this important point. As the Reviewer suggested, we added some of the missing information about CXCR1/2 expression in TME cells, their downstream pathways and biological effects, in the revised Introduction section (revised paragraph 1). In particular, we have better described the role of IL-8 in angiogenesis and cancer stem cells maintenance, as well as its involvement in macrophage recruitment.

It is challenging to penetrate all the results and get an overview of this manuscript. In addition, many results are presented in supplement which makes it even harder to get the overview.

We thank the Reviewer for highlighting this point. As the Reviewer suggested, we have now moved some Supplementary Figures into the main manuscript, as follows:

  • Supplementary Figure 5A and B have been moved into the main manuscript as revised Figure 6 (panels D and E)
  • Supplementary Figure 6A and B have been moved as revised Figures 7A and B

(see also general editorial comments above).

REVIEWER #3:

The article on the role of IL8 in CRC was of great interest and written in an organized concise way and very comprehensive. Actually, it exceeds my expectation while going into the details.

We thank the Reviewer for his/her recognition of the relevance of our report.

In section 2.3 (methodology): How the percentage was assessed? Did authors evaluate cellular location?

We thank the Reviewer for highlighting this important point. The intensity of both PTEN and IL-8 staining was detected in the cytoplasm and assessed according to a standardized score used in several papers from our group (see, for example, Pino MS et al., 2008). Briefly, PTEN and IL-8 levels in tumor cells were evaluated by a standardized scoring system (0 = no staining or weak cytoplasmic staining in <10% of cells, 1+ = faint/barely perceptible cytoplasmic staining in >10% of cells, 2+ = weak to moderate cytoplasmic staining in >10% of cells, 3+ = strong cytoplasmic staining in >10% of cells). For infiltrating immune cells, IL-8 was assessed in extravascular neutrophils and macrophages, using the same scoring system. This is now described in more detail in revised paragraph 2.3, lines 156-161). Except for a generic assessment of cytoplasmic staining (as reported), it was not possible, by IHC, to localize positive staining to specific subcellular structures.

A bit more expansion for sequencing in section2.4 is important. I know there is a citation, but simple description will be useful to the reader. Was it whole or targeted?

We thank the Reviewer for highlighting this point. We have now added further information on the Ion Ampliseq Cancer Hotspot Panel and Ion S5 Sequencer used in the revised paragraph 2.4 (lines 187-196). Further details regarding the NGS method, and data analysis and reporting are reported in Conciatori F et al., 2020 (ref. 17 of the revised manuscript).

REVIEWER #4:

The Article review titled as “Interleukin-8 in colorectal cancer: A systematic review and meta-analyses of its potential role as a prognostic biomarker” may be publishable, but it should be reviewed after minor revisions Comments: The review provides in depth information.

The review provides in depth information, the article is well written; the references are appropriate. Therefore, its publication in “Biomedicines” is recommended after minor revision.

We thank the Reviewer for his/her recognition of the relevance of our report.

Page No, 1, Line 32, replace “iiii)” with “iv)”

Page No. 3, in Table 1, Adjust the border line.

Typos have now been corrected in the revised version of the manuscript.

Page No. 5, Line 202-204, “cite the relevant paper”

We thank the Reviewer for highlighting this point. We added relevant references on IL-8 regulation.

In reference section please keep the journal writing style (as per Biomedicines) CONSISTENT for all the listed references.

Text size throughout the review is not consistent, please correct accordingly

See general editorial comments above.

On behalf of all authors we would like to take this occasion to thank the reviewers once again for their suggestions, which have certainly to a substantial improvement of the revised manuscript, and look forward to the final editorial decision.

Yours sincerely,

Chiara Bazzichetto and Michele Milella

Chiara Bazzichetto, PhD                                                     

Preclinical Models and New Therapeutic Agents

IRCCS - Regina Elena National Cancer Institute

Via Elio Chianesi, 53                                                   

00144 Rome, Italy                                                       

E-mail: chiara.bazzichetto@ifo.gov.it

Michele Milella, MD, PhD

Section of Oncology, Department of Medicine

University of Verona and Verona University Hospital

P.le L.A. Scuro, 10

37134 Verona, Italy

E-mail: michele.milella@univr.it

Reviewer 2 Report

The study by Bazzichetto et al, “Interleukin-8 in colorectal cancer: A systematic review and 2 meta-analysis of its potential role as a prognostic biomarker.” is a combination of a) literature review 2) meta-analysis c) database transcriptome analysis 4) CRC sample analysis

I have the following comments:               

-The combination of 1-4 above give a confusing impression, is complex and makes it very hard to penetrate and review the manuscript.

-The Introduction is lacking a background about the cytokine IL-8 in normal and CRC cells: function, role and cell types producing IL-8 (both intracellular and soluble), cell types expressing receptors and the effect of IL-8 ligation through CXCR1/2. The introduction is also lacking the role of the different signalling mechanisms involved in CXCR1/2 pathways.

It is challenging to penetrate all the results and get an overview of this manuscript. In addition, many results are presented in supplement which makes it even harder to get the overview.

I ca not see that this paper comes up with any other results than those published by Xia et al 2015, as mentioned on line 69-70, A meta-analysis conducted by Xia et al. in 2015 suggested a role for IL-8 expression in CRC diagnosis, staging, and prognosis”

I suggest that the authors separate the literature study from their own original data and try to write two papers instead.

Author Response

(The authors gave the same response as above.)

Reviewer 3 Report

The article on the role of IL8 in CRC was of great interest and written in an organized concise way and very comprehensive. Actually, it exceeds my expectation while going into the details.

In section 2.3 (methodology): How the percentage was assessed? Did authors evaluated cellular location?

A bit more expansion for sequencing in section2.4 is important. I know there is a citation, but simple description will be useful to the reader. Was it whole or targeted?

Author Response

(The authors gave the same response as above.)

Author Response

(The authors gave the same response as above.)

Round 2

Reviewer 2 Report

Revisions have been done according to reviewers. I am satisfied with the revised manuscript.